# Application of *Lactobacillus reuteri* B1/1 (*Limosilactobacillus reuteri*) Improves Immunological Profile of the Non-Carcinogenic Porcine-Derived Enterocytes

**DOI:** 10.3390/life13051090

**Published:** 2023-04-27

**Authors:** Viera Karaffová, Jana Teleky, Maša Pintarič, Tomaž Langerholc, Dagmar Mudroňová, Erik Hudec, Zuzana Ševčíková

**Affiliations:** 1Department of Morphological Disciplines, University of Veterinary Medicine and Pharmacy in Košice, Komenského 73, 04181 Košice, Slovakia; jana.teleky@uvlf.sk (J.T.); erik.hudec@uvlf.sk (E.H.);; 2Department of Microbiology, Biochemistry, Molecular Biology and Biotechnology, Faculty of Agriculture and Life Sciences, University of Maribor, Pivola 10, 2311 Hoče, Slovenia; masa.pintaric@um.si (M.P.); tomaz.langerholc@um.si (T.L.); 3Department of Microbiology and Immunology, University of Veterinary Medicine and Pharmacy in Košice, Komenského 73, 04181 Košice, Slovakia; dagmar.mudronova@uvlf.sk

**Keywords:** *Lactobacillus reuteri* B1/1, CLAB cells, pro-inflammatory cytokine, metabolic activity

## Abstract

In our previous studies, *Lactobacillus reuteri* B1/1, which was renamed *Limosilactobacillus reuteri* (*L. reuteri*), was able to modulate the production of pro-inflammatory cytokines and other components of the innate immune response in vitro and in vivo. In this study, we evaluated the effect of *Lactobacillus reuteri* B1/1 in two concentrations (1 × 10^7^ and 1 × 10^9^ CFU) on the metabolic activity, adherence ability and relative gene expression of pro-inflammatory interleukins (IL-1β, IL-6, IL-8, IL-18), lumican and olfactomedin 4 produced by non-carcinogenic porcine-derived enterocytes (CLAB). CLAB cells were cultured in a 12-well cell culture plate at a concentration of 4 × 10^5^ cells/well in DMEM medium in a controlled humidified atmosphere for 48 h. A 1 mL volume of each probiotic bacterial suspension was added to the CLAB cells. Plates were incubated for 2 h and 4 h. Our results revealed that *L. reuteri* B1/1 was able to adhere to CLAB cells in sufficient numbers in both concentrations. In particular, the concentration of 10^9^ *L. reuteri* B1/1 allowed to modulate the gene expression of pro-inflammatory cytokines, as well as to increase the metabolic activity of the cells. In addition, administration of *L. reuteri* B1/1 in both concentrations significantly stimulated gene expression for both proteins in the CLAB cell line after 4 h of incubation.

## 1. Introduction

The extensive breeding of farm animals and the demands of consumers on the quality of animal products place emphasis on breeders and producers in the direction of improving animal health. In this context, in the last decade, the attention of the public and scientists has been focused on gaining knowledge about the use of probiotic bacteria and their products in animal health and diseases [1].

However, the range of probiotic properties varies considerably between strains belonging to different genera and species. The most common bacterial strains used as probiotics can be found among the lactic acid bacteria (LAB) species, with the *Lactobacillus* genus being the most studied. Recent studies also show the fact that the most promising probiotic species appear to be those derived from the genus *Lactobacillus* [2,3]. Due to phenotypic, genotypic and ecological diversity, the genus *Lactobacillus*, which currently consists of more than 261 species, was reclassified in 2020 into 25 genera. The latter includes the genus *Lactobacillus*, which includes host-adapted organisms and 23 new genera, which include *Holzapfelia, Lapidilactobacillus, Amylolactobacillus, Bombilactobacillus* and others [4].

Several members of the *Lactobacillus* genus have claimed “generally recognized as safe” (GRAS) status, confirming their safety [5]. In particular, *Lactobacillus reuteri* recently renamed *Limosilactobacillus reuteri* (*L. reuteri*) is the most studied probiotic strain that colonizes the intestine of a large number of mammals and humans [6]. It has been proven that *L. reuteri* produces some antimicrobial substances such as exopolysaccharides and is therefore able to reduce pathogenic colonization. At the same time, it influences the composition of the host commensal intestinal microbiota. *L. reuteri* can modulate the production of pro-inflammatory and anti-inflammatory cytokines while supporting the development and function of T cells [7]. Additionally, *L. reuteri* colonization strengthens the intestinal barrier, which may reduce microbial translocation from the intestinal lumen to other host tissues, which usually initiates an inflammatory response [8,9].

*L. reuteri* B1/1, which was isolated from the intestine of pheasants, represents a promising candidate in terms of its use in commercial probiotic preparations for poultry. This hypothesis is supported by results from recent studies where *L. reuteri* B1/1 was able to stimulate the formation of NLRP3 inflammasome and co-operating immune molecules in the cecum of broiler chickens during *Campylobacter jejuni* infection [10]. These results were also confirmed in our previous in vitro study, when *L. reuteri* B1/1 effectively modulated the gene expression of pro-inflammatory cytokines (IL-1β, IL-15), chemokine (MIP-1β) and the percentage of T-cell subpopulations (CD3+, CD4+ and CD8+) in blood mononuclear cells (PMBCs) isolated from poultry peripheral blood [11].

However, the integrity of the intestinal mucosa and mucosal immunity is essential to create a suitable environment not only for the absorption of nutrients and protection against the invasion of intestinal pathogens, but also for the microbiota. Glycoprotein OLFM4 and proteoglycan lumican have a demonstrably important role, mainly in innate immunity against pathogens of bacterial origin and in inflammatory diseases of the digestive tract and some types of cancer [12,13]. Moreover, in recent years of research, it has been shown that both proteins are necessary for ensuring the integrity of the mucosa mainly during infection and wound healing [14]. Likewise, TLR signaling in the intestine is markedly involved in the regulation of immune tolerance to commensals, as well as host immune reaction to the presence of intestinal pathogens. In this regard, the epithelial expression of TLR4, which recognizes lipopolysaccharide of Gram-negative bacteria, and TLR5, which recognizes bacterial flagellum, have a strategic role in the co-operation of their signaling pathways during the recognition of bacterial pathogens by the innate immune system [15,16].

For a deeper understanding of the mechanism of action of *L. reuteri* B1/1 on host mucosal immunity, further studies are necessary. Therefore, the aim of this study was to investigate the effect of *Lactobacillus reuteri* B1/1 at two concentrations (10^7^ and 10^9^ CFU/mL) on metabolic activity, adherence ability, relative gene expression of interleukins (IL-1β, IL-6, IL-8, IL-18) and important proteins during the recovery of damaged epithelium (lumican proteoglycan, olfactomedin 4 glycoprotein) produced by non-carcinogenic porcine-derived enterocytes (CLAB).

## 2. Materials and Methods

### 2.1. Cell Culture

#### 2.1.1. Growth and Maintenance

Non-carcinogenic porcine-derived enterocytes cell line (CLAB) was developed in the Department of Biochemistry and Nutrition of the Faculty of Medicine and in the Department of Microbiology, Biochemistry, Molecular Biology and Biotechnology of the Faculty of Agriculture and Life Sciences of the University of Maribor. It consists of epithelial cells from the small intestine (cecum) of an adult pig [17]. CLAB cells were grown in Dulbecco’s Modified Eagle’s (DMEM) advanced medium (Life Technologies, Carlsbad, CA, USA), supplemented with 100 IU/mL penicillin and 0.1 mg/mL streptomycin (both from Sigma-Aldrich, Saint Louis, MO, USA), 2 mM L-glutamine and 5% fetal bovine serum (FBS) (both from Life Technologies). Cells were incubated in 25 cm^2^ cell culture flasks (Corning Inc., Corning, NY, USA) under controlled humidified conditions (37 °C, 5% CO_2_). Medium was changed as needed, and cells were passaged several times before use in the experiments.

#### 2.1.2. Bacterial Strain Growth and Maintenance

The *Lactobacillus* strain used in the present study was isolated from the intestinal contents of healthy pheasants. The strain was identified as *Lactobacillus reuteri* B1/1 (*L. reuteri* B1/1) by matrix-assisted laser desorption/ionization–time of flight mass spectrometry (MALDI–TOF MS) at the Centre of Biosciences of the Slovak Academy of the Institute of Animal Physiology in Košice [18]. Stock cultures of *L. reuteri* B1/1 and *Lactobacillus rhamnosus* GG (LGG) (ATCC 53103) as control were maintained at −80 °C in De Man Rogosa, Sharpe (MRS) broth (Merck kGaA, Darmstadt, Germany) containing 20% (*v*/*v*) glycerol (Merck kGaA). Both strains were propagated at least 3 times in MRS broth (Merck kGaA) for 24 h at 35 °C in a CO_2_-enriched atmosphere according to the manufacturer’s instructions before use in the experiments.

### 2.2. Viability of the Cell Line CLAB after Exposure to Probiotic Bacteria (MTT Assay)

Cell viability after exposure to the probiotic strain *L. reuteri* B1/1 was checked by assay to determine the metabolic activity of the CLAB cell line. CLAB were cultured in DMEM medium (as previously described) at a concentration of 5 × 10^4^ cells/well in a 96-well flat-bottomed microtiter plate (Corning™ Costar™, Thermo Scientific, Waltham, MA, USA) in a humidified atmosphere with 5% CO_2_ at 37 °C. Growth of the probiotic strain followed the same procedure as previously described. Once the cells were confluent, medium was removed from the cells, and they were washed once with sterile DMEM medium (200 µL) supplemented with 2 mM L-glutamine (both Life Technologies). The suspension of probiotic strain *L. reuteri* B1/1 in MRS broth (Merck) was centrifuged at 2400 rpm for 10 min. After centrifugation, the probiotic strain was resuspended in DMEM medium supplemented with 2 mM L-glutamine (both Life Technologies). A 100 µL volume of the bacterial suspension with a concentration of 10^7^ CFU/mL and 10^9^ CFU/mL was added to the cells. Cells with probiotics were incubated for 2 h and 4 h in a controlled atmosphere (5% CO_2_ at 37 °C). Combinations of CLAB with probiotic strain were prepared in three independent replicates, and three independent replicates of controls for each cell line were incubated in parallel with the same medium DMEM (Life Technologies) (100 µL) but without added probiotic strain.

Metabolic activity of the exposed cells was measured using the MTT (3-(4,5 dimethylthiazol-2-yl)-2,5-diphenyltetrazolium bromide—Sigma, St. Louis, MO, USA) assay. After incubation, cells were washed twice with sterile DMEM medium (200 µL) supplemented with 2 mM L-glutamine (both Life Technologies). Subsequently, 200 µL of MTT solution (Sigma)/well (final concentration: 0.5 mg/mL MTT in DMEM medium supplemented with 2 mM L-glutamine (both Life Technologies), 100 IU/mL penicillin and 0.1 mg/mL streptomycin (both Sigma) were added to the cells. The plate was placed on an orbital horizontal mixer for 5 min and then incubated in a humidified atmosphere with 5% CO_2_ at 37 °C for 2 h. After incubation, the medium with added MTT solution was discarded. The plate was allowed to dry completely at room temperature, and then 100 µL of 0.04% HCl (Sigma) in isopropanol (Sigma) was added to each well. The plate was incubated again for 5 min on an orbital horizontal mixer. A further 15 min incubation at room temperature was used to dissolve the formazan crystals and thus cause color formation. Absorbance was measured at OD 570 nm and 630 nm using a spectrophotometer M1000 PRO (Tecan, Trading AG, Männedorf, Switzerland). The final absorbance was calculated from the difference between the absorbance measurement at OD 630 nm and OD 570 nm. The absorbance values of the wells with added probiotic strains were expressed as a percentage (%) relative to the absorbance values in the control wells, which were set to 100%.

### 2.3. Adhesion Assay of L. reuteri B1/1 and L. rhamnosus GG (LGG) to CLAB Cells

To determine the ability of the probiotic to adhere to epithelial CLAB cells, the adhesion assay was performed as previously described [19] with slight modifications. A 0.1 mL volume of *L. reuteri* and LGG (10^7^ CFU/mL and 10^9^ CFU/mL) was added to a 96-well flat-bottomed microtiter plate (Corning™ Costar™, Thermo Scientific, USA) containing fully differentiated CLAB cells. To allow attachment, cells were incubated for 2 h with both probiotics in a controlled atmosphere (5% CO_2_ at 37 °C). Combinations of CLAB with probiotic strains were prepared in three independent replicates. Cells were then washed three times with warm phosphate-buffered saline (PBS) buffer (Sigma-Aldrich) and incubated with 0.5% Triton-X (Sigma) in PBS for 30 min at room temperature. Triton-X lyses the cell membrane, causing the probiotic to detach from the CLAB cells. A series of serial dilutions were then prepared in 0.9% saline solution (Sigma) and spread on MRS agar plates. After incubation at 30 °C for 72 h, colonies of the two probiotic strains were counted. The results are expressed as the number of bacteria adhered to 1 cm^2^ of the gut.

### 2.4. Design of the Experiment

Once confluent, cell monolayers were washed twice with warm phosphate-buffered saline (PBS) buffer (Sigma-Aldrich) to remove antibiotics. Then, CLAB cells were cultured in a 12-well cell culture plate (Corning Incorporated) at a concentration of 4 × 10^5^ cells/well in DMEM medium containing P/S and L-glutamine (1.5 mL) in a controlled humidified atmosphere with 5% CO_2_ at 37% for 48 h. The probiotic strain *L. reuteri* B1/1 in MRS broth (Merck KGaA) was centrifuged at 2400 RPM for 10 min and resuspended in DMEM advanced media supplemented with 2 mM L-glutamine (both from Life Technologies) and without antibiotics. A 1 mL volume of the probiotic bacterial suspension at a concentration of 10^7^ (LR7) and 10^9^ (LR9) CFU/mL was added to the CLAB cells. As a control (C), the same medium but without bacteria was used for cell exposure. Plates were incubated for 2 h and 4 h under a controlled atmosphere (5% CO_2_ at 37%). All treatments and controls were performed in three independent replicates. After incubation, the cells were immediately washed with ice-cold sterile PBS buffer (Sigma-Aldrich).

### 2.5. Measurement of Gene Expression

#### 2.5.1. Isolation of Total RNA from Cell Culture

A 500 μL volume of TRI-reagent/well (Sigma-Aldrich) was added to cell monolayers and was collected from each well by pipetting in order to extract RNA. For the purification of total RNA, the RNeasy mini kit was used (Qiagen, Manchester, UK), according to the manufacturer’s instructions. Concentration and purity of total RNA were determined spectrophotometrically on NanoPhotometer P-Class P 300 (Implen, München, Germany) at ratios of 260/280 and 260/230. For all isolated RNA samples, the ratio was 260/280 around 2.0, and the 260/230 ratio was around 1.8 to 2.2, indicating minimal contamination. Subsequently, 1 μg of the total RNA was reverse transcribed using the iScript cDNA Synthesis Kit (Bio-Rad, Hercules, CA, USA), as described by Karaffová et al. [20].

#### 2.5.2. Quantitative Real-Time PCR

The relative gene expression of interleukins (IL-1β, IL-6, IL-8, IL-18), TLR5 and proteins (OLFM4, LUM) was determined. In addition, mRNA relative expression of the reference gene encoding hypoxanthine-guanine phosphoribosyltransferase (HPRT) was determined based on the stability of expression using geNorm software. Primer sequences used for qRT-PCR are shown in Table 1. All primer sets allowed DNA amplification efficiencies between 94% and 100%.

Changes in transcription and relative expression of selected genes were measured by real-time PCR and performed on the Lightcycler 480 II Instrument (Roche) using SsoAdvancedTM Universal SYBR Green Supermix (Bio-Rad Laboratories, Hercules, CA, USA) and specific primers (Table 1). Primers for the amplification of OLFM4 and LUM genes were designed using the gene sequences of OLFM4 (XM_003482903.4) and LUM (NM_001243339.1) from the GenBank database and using Primer-3 software, version 9.

The reaction mixture contained 10 μL of master mix, 7.5 μL of primers and 2.5 μL of the sample cDNA. The qRT-PCR reaction was initiated by denaturation at 95 °C for 30 s, followed by 37 cycles of amplification–denaturation at 95 °C for 15 s, annealing at 60 °C for 30 s and an elongation step at 72 °C for 2 min. A melting curve from 65 °C to 95 °C with readings at every 0.5 °C was noticed for each individual RT-qPCR plate. Analysis was performed after every run to ensure a single amplified product for each reaction. Each real-time PCR reaction was performed in duplicate, and mean values of the duplicate were used for further analysis. We confirmed that the amplification efficiency of each gene (including HPRT) was between 94% and 100% in the exponential phase of the reaction, where the quantification cycle (Cq) was calculated. The Cq values of the genes studied were normalized to the average Cq value of the reference gene (ΔCq), and the relative expression of each gene was calculated mathematically as 2^−ΔCq^.

### 2.6. Statistical Analyses

#### 2.6.1. Viability of Cell Line CLAB after Exposure to Probiotic Bacteria (MTT Assay)

Obtained data were analyzed with SPSS IBM Statistics 24.0 (IBM Inc., Armonk, NY, USA). Data were first analyzed using the Shapiro–Wilk normality test. Subsequently, analyses were carried out using the Mann–Whitney U-test. α level was set to 5%, and * *p* < 0.05 was considered statistically significant.

#### 2.6.2. Relative Gene Expression

Obtained results were evaluated using GraphPad Prism 9.0.0 software (GraphPad Software, version 9, San Diego, CA, USA) by the Shapiro–Wilk normality test, followed by one-way analysis of variance (ANOVA). Values in figures are given as means resp. medians in the case of relative gene expression with standard deviations (± SD).

## 3. Results

### 3.1. Viability of the Cell Line CLAB after Exposure to Probiotic Bacteria (MTT Assay)

Before testing the effect of the probiotic bacteria on the CLAB cell line, the viability of the cells was measured after 2 h and 4 h exposure to the probiotic strain *L. reuteri* B1/1 with a concentration of 10^7^ CFU/mL (LR7) and 10^9^ CFU/mL (LR9). Incubation of the cell line with the probiotic strain showed increased metabolic activity of the cells and consequently increased cell viability. The addition of the probiotic strain LR7 increased the metabolic activity of the cell by 4% and by 0.2% after 2 h and 4 h, respectively, but without statistical difference compared to the control. However, probiotic strain LR9 increased the metabolic activity of the CLAB cell line statistically significantly by 87% and 118% after 2 h and 4 h, respectively, compared to the control (both *p < 0.05*) (Figure 1).

Viability of the cell line CLAB after exposure to *Lactobacillus reuteri* B1/1 (10^7^ and 10^9^) (MTT assay).

### 3.2. Adhesion Assay

After the addition of *L. reuteri* B1/1 at a concentration of 1 × 10^7^ CFU/mL media, which represents 5.95 × 10^6^ bacteria per 1 cm^2^ of the gut, 4 × 10^5^ bacteria per 1 cm^2^ of the gut adhered to the CLAB cells. *L. reuteri* B1/1 at a concentration of 1 × 10^9^ CFU/mL media, which represents 5.95 × 10^8^ bacteria per 1 cm^2^ of the gut, adhered to the CLAB cells 3 × 10^6^ bacteria per 1 cm^2^ of the gut. On the other hand, the control strain LGG at a concentration of 1 × 10^7^ CFU was able to adhere to CLAB cells in the number of 2.6 × 10^5^ bacteria, while at a concentration of 10^9^ CFU, the adhesion capacity was 2.2 × 10^6^ bacteria/cm^2^ of the gut.

### 3.3. Relative Expression for Selected Genes

Relative expression for the IL-1β gene was upregulated in both *L. reuteri* B1/1 groups after 2 h of incubation in comparison with the control (*p* < 0.01; *p* < 0.0001). The same situation was recorded after 4 h of incubation (*p* < 0.01; *p* < 0.001) (Figure 2).

Likewise, relative expression for the IL-6 gene was upregulated in both lactobacilli groups compared to the control in 2 and in 4 h of incubation (*p* < 0.01; *p* < 0.001; *p* < 0.0001). The highest level was recorded in the LR9 group in comparison with the control (*p* < 0.001) after 4 h (Figure 3).

The level of relative expression for the IL-8 gene was the highest at a 10^9^ concentration of *L. reuteri* B1/1 compared to LR7 (*p* < 0.01) and the control (*p* < 0.0001) after 2 h of incubation. Similar results were observed in the same group compared to the LR7 (*p* < 0.05) and the control (*p* < 0.0001) after 4 h (Figure 4).

Relative expression for the IL-18 gene was upregulated in both LR groups compared to the control (*p* < 0.05; *p* < 0.001) after 2 h of incubation. After 4 h of incubation, the highest measured relative gene expression was in the LR9 group compared to LR7 (*p* < 0.05) and the control (*p* < 0.0001) (Figure 5).

The relative expression for the TLR4 gene was markedly upregulated in the LR7 group compared to the control (*p* < 0.05) after 2 h of incubation. After 4 h of incubation, the highest level of expression was recorded in the LR9 group in comparison with other groups (*p* < 0.05) (Figure 6).

The highest level of relative expression for TLR5 was observed in the LR7 group in comparison with LR9 (*p* < 0.001) and control (*p* < 0.0001) after 2 h of incubation. After 4 h of incubation, there was a significant increase in gene expression in the LR9 group compared to the control (*p* < 0.001). On the contrary, in LR7, there was a decrease in the level of gene expression compared to the 2 h incubation, but it was still significantly higher than the control (*p* < 0.001) (Figure 7).

Relative expression for both proteins (OLFM4 and LUM) was markedly upregulated in lactobacilli groups only after 4-h of incubation in comparison with control (*p* < 0.05; *p* < 0.01; *p* < 0.001) (Figure 8 and Figure 9).

## 4. Discussion

The addition of probiotic bacteria as functional food supplements has become popular due to their many health benefits. According to animal studies and preclinical results, it was proven that the genus *Lactobacillus* may help in the prevention and treatment of many gastrointestinal infections and diseases [25]. On the other hand, the development of each new probiotic preparation includes research on the selected production strain, where it is necessary, among other things, to have detailed knowledge of the mechanism of its action on the host [26].

Our results showed that the incubation of the CLAB cell line with probiotic bacteria significantly increased the metabolic activity of the cells and subsequently increased the cell viability of the CLAB cell line markedly at a concentration of 10^9^ CFU/mL after both exposure times. The data obtained in this study indicate that the response of CLAB cells to *Lactobacillus reuteri* B1/1 exposure varies depending on its concentration and duration of exposure.

A similar study was conducted by Castiblanco et al. [27], where the aim of their in vitro study was testing the cell viability of human gingival fibroblast (HGF) and its production of prostaglandin E2 (PGE2) when exposed to supernatants of two mixed *Lactobacillus reuteri* strains (ATCC PTA 5289 and DSM 17938). Their findings showed that none of the *L. reuteri* supernatants were cytotoxic and affected the viability of HGF. These results may suggest that none of the bacterial components of our used strain *Lactobacillus reuteri* B1/1 are harmful to porcine enterocytes.

Our obtained results revealed a higher ability of *L. reuteri* B1/1 to adhere to CLAB cells in both concentrations in comparison with the control LGG strain. While the ability to adhere to intestinal epithelial cells is considered a basic prerequisite for the successful action of LAB probiotic strains. At the same time, a high adhesive ability can support the residence time of LAB strains in the intestine, limit potential pathogens and thus protect intestinal epithelial cells [28]. In this meaning, Šikić Pogačar et al. [29] found that a mixture of lactobacilli strains (PCS20, PCS22, PCS25, LGG, PCK9) successfully reduced the adhesion ability of *Campylobacter jejuni* under co-culture conditions on CLAB cells.

Epithelial cells, including intestinal epithelial cells, are known to be an effective physical barrier. For this function, they have evolved innate immune antimicrobial functions and the ability to modulate immune cell recruitment and activity during infection. By secreting chemokines such as IL-8, epithelial cells direct immune cells to sites of infection and simultaneously secrete other cytokines IL-6 and IL-1β to activate inflammatory immune responses of both the innate and adaptive immune systems [30].

Pro-inflammatory cytokines, such as IL-1β and IL-6, are mainly produced by macrophages, monocytes and by non-immune cells during pathogen invasion, cell injury and especially in the initial stage of inflammation [31]. Another pro-inflammatory cytokine IL-18 is involved together with the other aforementioned interleukins and cytokines in the signaling pathway of many inflammatory diseases. It also regulates the innate and acquired immune response and mature IL-18 induces IL-8 [32]. Neutrophil chemotactic factor IL-8, which is expressed in a variety of cell types, mediates its biological effects through binding to its cognate G-protein-coupled CXC chemokine receptors, CXCR1 and CXCR2, which activate a phosphorylation cascade that initiates neutrophil chemotaxis as part of the inflammatory response [33].

On the other hand, it is known that excessive production of pro-inflammatory cytokines in the intestine has been shown to have a destructive effect on intestinal tissue [34]. In this regard, it is important to modulate immune reactivity mainly during infection. An important fact is that precisely probiotic bacteria are able to regulate the production of cytokines. In this study, gene expression of pro-inflammatory interleukins was markedly upregulated in both lactobacilli groups. We assume that this is a normal reaction of CLAB cells to the presence of stimulatory factors, without internal control mechanisms that are normally present in the host organism. In agreement with this statement, we confirmed in our previous study that *L. reuteri* B1/1 was able to modulate the production of pro-inflammatory cytokines and other components of innate immune reaction during campylobacteriosis in broilers in a desirable direction [10]. Similarly, in our previous in vitro study, strain *L. reuteri* B1/1 stimulated the relative gene expression for pro-inflammatory interleukins (IL-1β, IL-15) the most of all strains tested [11]. Moreover, co-operation between interleukin signaling and the recognition of bacterial components by TLR receptors is essential for the control of many bacterial infections, while it is generally known that TLR signaling pathways culminate in the transcriptional activation of nuclear factor kappaB (NF-κB), which regulates the expression of a number of pro-inflammatory genes, including cytokines, chemokines and adhesion molecules [35,36,37]. Akhtar et al. [38] reported in their study that human intestinal epithelial cells also increase the expression of IL-6 and IL-8 after exposure to CpG and LPS bacteria, which were recognized by TLR9 and TLR4. The *Lactobacillus* strain we used significantly stimulated gene expression for TLR4 and TLR5 in both concentrations after 2 h of incubation. Likewise, an in vivo study on the intestine of broiler chickens revealed the ability of *L. reuteri* at a concentration of 2 × 10^9^ CFU/mL to effectively modulate the expression of innate immune molecules, including TLRs [39]. Additionally, Alizadeh et al. [40] observed that *in ovo* administration of lactobacilli mix at a dose of 10^7^ CFU enhanced antibody responses by increasing the percentage of CD4+ and CD4+CD25+ T cells and upregulating pro-inflammatory cytokines in the spleen and bursa. Overall, our results suggest that the application of *L. reuteri* B1/1 at the two concentrations used may be crucial for generating robust immune responses. Moreover, we suppose that the levels of pro-inflammatory factors are just enough uploaded and therefore may increase the level of immune system readiness for the invasion of pathogens.

However, the integrity of the intestinal wall is a fundamental factor in maintaining internal homeostasis, which creates favorable conditions for generating a sufficient immune response to the presence of pathogens. OLFM4 and LUM are involved in the maintenance of the integrity of the intestinal epithelial layer and may interact with integrin receptors of epithelium during healing epithelial defects [41]. Another human study confirmed that these proteins are potential biomarkers also for evaluating the state of intestinal maturation [42].

Through the development and maintenance of a diverse and healthy commensal microbiota in the intestine, the integrity of the intestinal epithelial layers is improved, which subsequently increases the mucosal defense capacity of the intestine. Additionally, PICRUST microbial community analysis found that probiotic supplementation can increase branched-chain amino acid biosynthesis and butyrate metabolism, which also improves gut health integrity [43]. In terms of this statement, Shin et al. [44] observed that the supplementation of *Lactobacillus plantarum* JDFM LP11 promoted the integrity of intestinal epithelial layers in weaned piglets. The administration of *L. reuteri* B1/1 on CLAB cells markedly stimulated gene expression for both proteins after 4 h of exposition, which may indicate that our tested probiotic strain may be involved in the regeneration and healing processes of the intestinal epithelial layer. In addition, this effect can be crucial especially during the invasion of pathogens, when there is also mechanical damage to the mucous membranes of the affected organs and tissues. In this respect, Wu et al. [45] investigated the ability of *Lactobacillus reuteri* D8 to modulate the maintenance and regeneration of intestinal mucosal epithelium on crypt organoids isolated from the intestinal epithelia of mice. The research results demonstrated that *L. reuteri* D8 was effective in maintaining the regeneration of the intestinal epithelium and also protected it during *Citrobacter rodentium*-induced intestinal inflammation. In terms of these observations, we hypothesize that the application of *L. reuteri* B1/1 could represent a promising alternative for the prevention and therapy of intestinal inflammation. In this context, however, it is necessary to monitor the effect of *L. reuteri* B1/1 on inflammation-induced intestinal epithelial cells.

## 5. Conclusions

Our results revealed that *L. reuteri* B1/1 was able to adhere to CLAB cells in sufficient numbers in both concentrations. In particular, the concentration of 10^9^ *L. reuteri* B1/1 allowed to modulate the gene expression of pro-inflammatory cytokines, as well as to increase the metabolic activity of the cells. In addition, administration of *L. reuteri* B1/1 in both concentrations significantly stimulated gene expression for both lumican and olfactomedin 4 in an intact CLAB cell line after 4 h of incubation. Therefore, we hypothesize that *L. reuteri* B1/1 may also participate in strengthening the integrity of the intestinal mucosa.

## Figures and Tables

**Figure 1 life-13-01090-f001:**
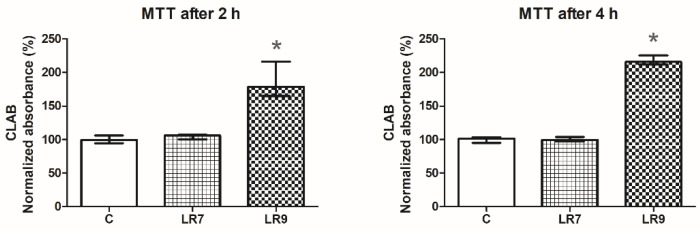
Viability test of CLAB upon co-culture with probiotic strain *L. reuteri* B1/1 with a concentration of 10^7^ CFU/mL (LR7) and 10^9^ CFU/mL (LR9). After 2 h and 4 h of incubation, viability was determined by MTT test and calculated relative to the control (C). Data from three independent experiments are presented as median (geometrical shapes) absorbances with range. (* *p* < 0.05).

**Figure 2 life-13-01090-f002:**
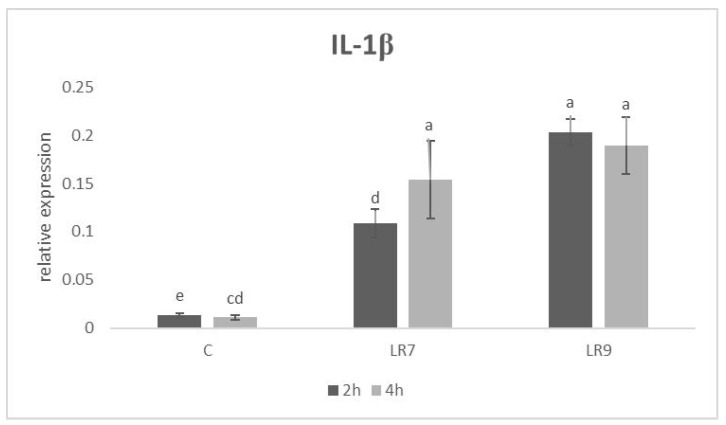
Relative expression of IL-1β gene in CLAB cells treated with *L. reuteri* B1/1 at a concentration of 10^7^ and 10^9^ CFU/mL. Results at each time point are the median of 2^−ΔCq^. Means with different superscripts are significantly different ^ac^
*p* < 0.01; ^ad^
*p* < 0.001; ^ae^
*p* < 0.0001. Legend: C—control, LR7—*L. reuteri* B1/1 (10^7^), LR9—*L. reuteri* B1/1 (10^9^).

**Figure 3 life-13-01090-f003:**
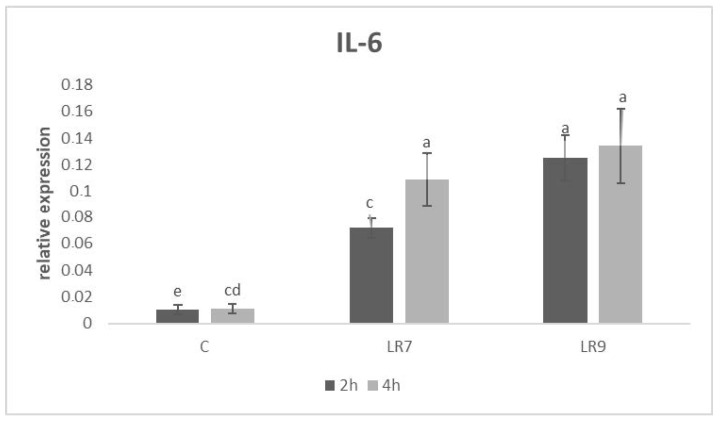
Relative expression of IL-6 gene in CLAB cells treated with *L. reuteri* B1/1 at a concentration of 10^7^ and 10^9^ CFU/mL. Results at each time point are the median of 2^−ΔCq^. Means with different superscripts are significantly different ^ac^
*p* < 0.01; ^ad^
*p* < 0.001; ^ae^
*p* < 0.0001. Legend: C—control, LR7—*L. reuteri* B1/1 (10^7^), LR9—*L. reuteri* B1/1 (10^9^).

**Figure 4 life-13-01090-f004:**
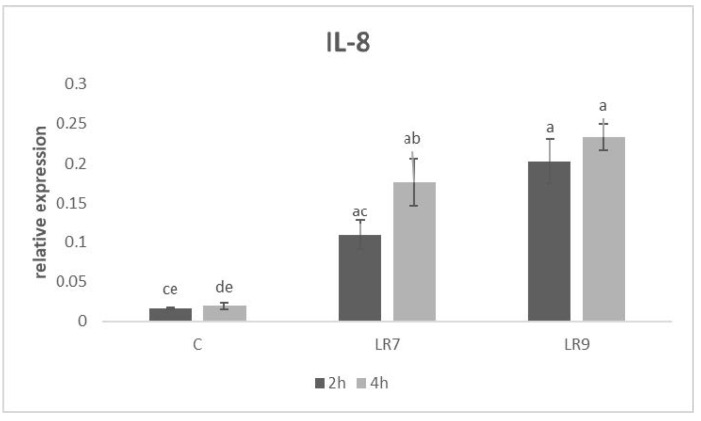
Relative expression of IL-8 gene in CLAB cells treated with *L. reuteri* B1/1 at a concentration of 10^7^ and 10^9^ CFU/mL. Results at each time point are the median of 2^−ΔCq^. Means with different superscripts are significantly different ^ab^
*p* < 0.05; ^ac^
*p* < 0.01; ^ad^
*p* < 0.001; ^ae^
*p* < 0.0001. Legend: C—control, LR7—*L. reuteri* B1/1 (10^7^), LR9—*L. reuteri* B1/1 (10^9^).

**Figure 5 life-13-01090-f005:**
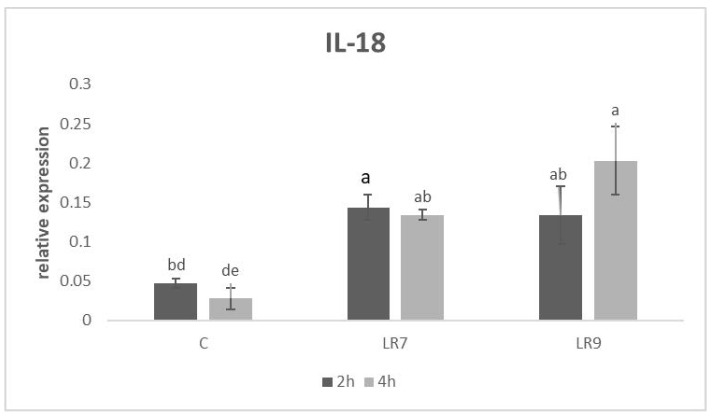
Relative expression of IL-18 gene in CLAB cells treated with *L. reuteri* B1/1 at a concentration of 10^7^ and 10^9^ CFU/mL. Results at each time point are the median of 2^−ΔCq^. Means with different superscripts are significantly different ^ab^
*p* < 0.05; ^ad^
*p* < 0.001; ^ae^
*p* < 0.0001. Legend: C—control, LR7—*L. reuteri* B1/1 (10^7^), LR9—*L. reuteri* B1/1 (10^9^).

**Figure 6 life-13-01090-f006:**
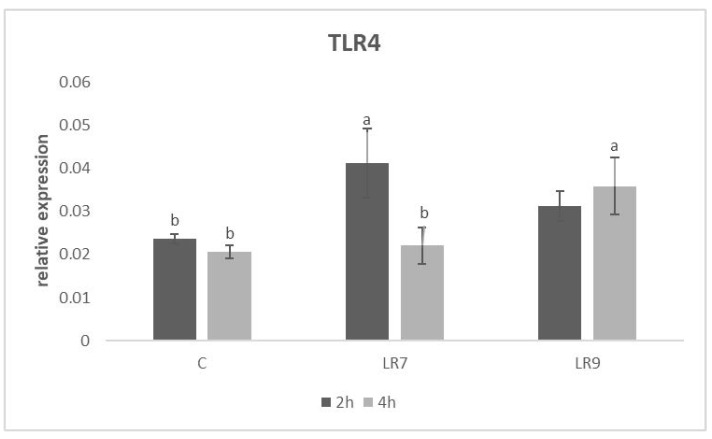
Relative expression of TLR4 gene in CLAB cells treated with *L. reuteri* B1/1 at a concentration of 10^7^ and 10^9^ CFU/mL. Results at each time point are the median of 2^−ΔCq^. Means with different superscripts are significantly different ^ab^
*p* < 0.05. Legend: C—control, LR7—*L. reuteri* B1/1 (10^7^), LR9—*L. reuteri* B1/1 (10^9^).

**Figure 7 life-13-01090-f007:**
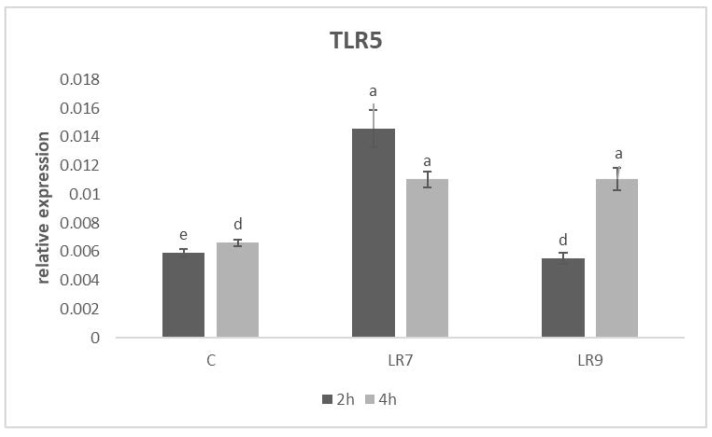
Relative expression of TLR5 gene in CLAB cells treated with *L. reuteri* B1/1 at a concentration of 10^7^ and 10^9^ CFU/mL. Results at each time point are the median of 2^−ΔCq^. Means with different superscripts are significantly different ^ad^
*p* < 0.001; ^ae^
*p* < 0.0001. Legend: C—control, LR7—*L. reuteri* B1/1 (10^7^), LR9—*L. reuteri* B1/1 (10^9^).

**Figure 8 life-13-01090-f008:**
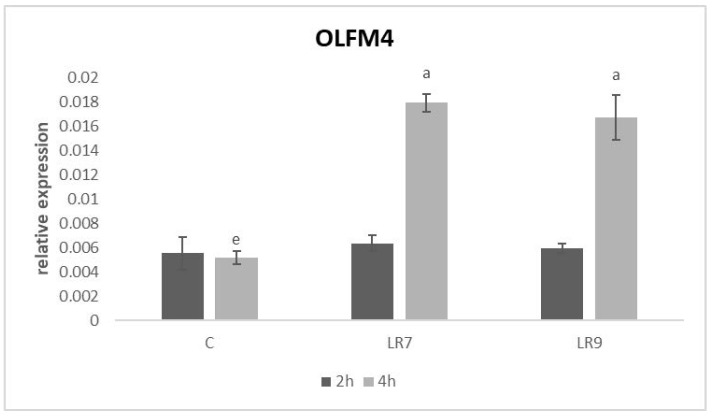
Relative expression of OLFM4 gene in CLAB cells treated with *L. reuteri* B1/1 at a concentration of 10^7^ and 10^9^ CFU/mL. Results at each time point are the median of 2^−ΔCq^. Means with different superscripts are significantly different ^ae^
*p* < 0.0001. Legend: C—control, LR7—*L. reuteri* B1/1 (10^7^), LR9—*L. reuteri* B1/1 (10^9^).

**Figure 9 life-13-01090-f009:**
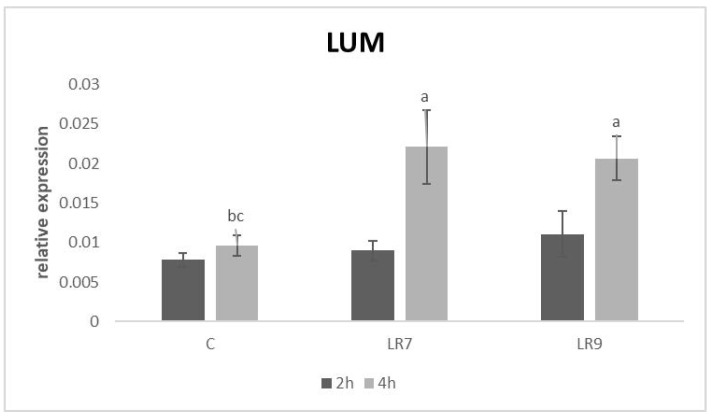
Relative expression of LUM gene in CLAB cells treated with *L. reuteri* B1/1 at a concentration of 10^7^ and 10^9^ CFU/mL. Results at each time point are the median of 2^−ΔCq^. Means with different superscripts are significantly different ^ab^
*p* < 0.05; ^ac^
*p* < 0.01. Legend: C—control, LR7—*L. reuteri* B1/1 (10^7^), LR9—*L. reuteri* B1/1 (10^9^).

**Table 1 life-13-01090-t001:** List of primer sequences used in qRT-PCR for target genes.

Primer	Sequence 5′–3′	References
IL-1β Fw	GAAGTGATGGCTAACTACGGTGAC	[21]
IL-1β Rev	ACCTGGACCTTGGTTCTCTGAGA	
IL-6 Fw	TGGGTTCAATCAGGAGACCT	[22]
IL-6 Rev	CAGCCTCGACATTTCCCTTA	
IL-8 Fw	TTATCGGAGGCCACAATAAG	
IL-8 Rev	TGGAATAGTAGATGGAGCCA	
IL-18 Fw	CTGCTGAACCGGAAGACAAT	[23]
IL-18 Rev	TCCGATTCCAGGTCTTCATC	
TLR4 Fw	CTCTGCCTTCACTACAGAGA	
TLR4 Rev	CTGAGTCGTCTCCAGAAGAT	
TLR5 Fw	TTTCTGGCAATGGCTGGACA	
TLR5 Rev	TGGAGGTTGTCAAGTCCATG	
OLFM4 Fw	GGTGATTTACGCAACTGAAG	In this study
OLFM4 Rev	GTTTGTACTGCTTGGTATGC	
LUM Fw	ACCTGCGTTTGTCTCATAAT	
LUM Rev	ATTGTAGGAGAGATCCAGCT	
HPRT Fw	AACCTTGCTTTCCTTGGTCA	[24]
HPRT Rev	TCAAGGGCATAGCCTACCAC	

## Data Availability

Not applicable.

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
