# Peer review of "Application of Lactobacillus reuteri B1/1 (Limosilactobacillus reuteri) Improves Immunological Profile of the Non-Carcinogenic Porcine-Derived Enterocytes"

_life, 2023, doi:10.3390/life13051090_

Round 1
Reviewer 1 Report
In this manuscript, the researchers tried to explain about the Application of Lactobacillus reuteri B1/1 (Limosilactobacillus reuteri) improves immunological profile of the non-carcinogenic porcine-derived enterocytes. It is interesting work and can be accepted after revision.
- The grammar errors should be checked in the whole manuscript.
- In abstract, the first four lines should be summarized.
- In introduction, the main objective has been repeated so it should be refined.
- Some recent and relevant articles may be added as thousands of articles have been published on this topic.
Please add graphical abstract as well to make a article attractive for reader
- Conclusion should be refined as it is not properly written as per results.
Author Response
Revision note
All changes in the manuscript are marked in yellow colour.
The grammar errors should be checked in the whole manuscript.
Thank you for your valuable comment, the grammar errors were checked in the whole manuscript.
In abstract, the first four lines should be summarized.
Thank you for your valuable comment, the first four lines was summarized.
In introduction, the main objective has been repeated so it should be refined.
Thank you for your valuable comment, but the main objective/aims of study are described in detail in the last sentence of the introduction (Line 82-87).
Some recent and relevant articles may be added as thousands of articles have been published on this topic.
Thanks for your opinion, but we think that our manuscript uses a sufficient number of current publications on this topic (at least the first six are highly current years 2022, 2020, 2019).
Please add graphical abstract as well to make a article attractive for reader
Thank you for your valuable comment. In this case, I would worry about the quality of the graphic abstract, because we have no experience with it. We would rather learn how to do it properly first and then put it into practice. Thank you for your understanding.
Conclusion should be refined as it is not properly written as per results.
Thank you for your valuable comment, the conclusion has been refined according to the main results.

Reviewer 2 Report
MDPI
Life-2328426
In this manuscript, the authors studied the effect of Lactobacillus reuteri B1/1 (L. reuteri B1/l) in two concentrations (1 x 107 and 1 x 109 CFU) on the metabolic activity, adhesive activity, and relative gene expression of pro-inflammatory interleukins (IL-1B, IL-6, IL-8, IL-16), lumican and olfactomedin 4 produced by noncarcinogenic porcine-derived enterocytes (CLAB). The authors concluded and reported that L. reuteri B1/l was able to adhere to CLAB cells in sufficient numbers. This is a well-designed and well-written manuscript. The authors have presented new information in the field of Lactobacilluis.
Author Response
Revision note
All changes in the manuscript are marked in yellow colour.
In this manuscript, the authors studied the effect of Lactobacillus reuteri B1/1 (L. reuteri B1/l) in two concentrations (1 x 107 and 1 x 109 CFU) on the metabolic activity, adhesive activity, and relative gene expression of pro-inflammatory interleukins (IL-1B, IL-6, IL-8, IL-16), lumican and olfactomedin 4 produced by noncarcinogenic porcine-derived enterocytes (CLAB). The authors concluded and reported that L. reuteri B1/l was able to adhere to CLAB cells in sufficient numbers. This is a well-designed and well-written manuscript. The authors have presented new information in the field of Lactobacilluis.
Dear reviewer from MDPI,
We would like to thank you very much for your comment and support of our research results. Your support is very important to us.
Kind regards,
Dr. Karaffová
Corresponding author

Reviewer 3 Report
This study is about effect of lactobacillus reuteri B1/1 on cell viability and gene expression of enterocyte cell line.
The results indicated that the lactobacillus increased cell viability and adherence to the cell line as well as altered inflammatory-related gene expressions.
The author concluded that the lactobacilius strain could help to enhance integrity of intestinal mucosa.
This MS is well written and understandable.
However, few things should be addressed before it is considered as a publication.
Abstract should be re-written with impactful sentences by mentioning more specific results.
line 64-66 and 385-386: be specific of the results. This may be imporatant to compare the effects of the L. reuteri on inflammatory-related gene expressions of blood and intestine (this study).
Discussion and Conclusion: as Author mentioned, L. reuteri may have anti-inflammatory roles that can strength intestinal mucosa.
For this, author needs to metion further studies on TLR-associated NF-kB signaling pathway, which is the pathway connected to production of inflammatory cytokines.
Author Response
Revision note
All changes in the manuscript are marked in yellow colour.
This study is about effect of lactobacillus reuteri B1/1 on cell viability and gene expression of enterocyte cell line.
The results indicated that the lactobacillus increased cell viability and adherence to the cell line as well as altered inflammatory-related gene expressions.
The author concluded that the lactobacilius strain could help to enhance integrity of intestinal mucosa.
This MS is well written and understandable.
However, few things should be addressed before it is considered as a publication.
Abstract should be re-written with impactful sentences by mentioning more specific results.
Thank you for your valuable comment, abstract was re-written with impactful sentences by mentioning more specific results.
line 64-66 and 385-386: be specific of the results. This may be imporatant to compare the effects of the L. reuteri on inflammatory-related gene expressions of blood and intestine (this study).
Thank you for your valuable comments, specific results received in our previous studies were added.
Discussion and Conclusion: as Author mentioned, L. reuteri may have anti-inflammatory roles that can strength intestinal mucosa. For this, author needs to metion further studies on TLR-associated NF-kB signaling pathway, which is the pathway connected to production of inflammatory cytokines.
Thank you for your valuable comment, further studies about TLR-associated NF-kB signaling pathway were added.
